# Contextual Factors and the Diffusion of MAIs in Manufacturing and Non-Manufacturing Sectors in Libya

Alhadi Boukr [1,*], Hassan Yazdifar [1] and Davood Askarany [2]

1. Department of Accounting, Finance and Economics, Bournemouth University, 89 Holdenhurst Road, Bournemouth BH8 8EB, UK; hyazdifar@bournemouth.ac.uk
2. Department of Accounting & Finance, Business School, University of Auckland, Auckland 92019, New Zealand; d.askarany@auckland.ac.nz
* Correspondence: alhadiboukr@gmail.com

**Abstract:** The diffusion of innovation theory has already addressed the major contextual factors hindering or facilitating the diffusion of management accounting innovations (MAIs) in organisations. However, the diffusion of MAIs in less developed countries (such as Libya) is still very low, and the contextual factors addressed by the diffusion of innovation seem to fail to explain the low diffusion. To address this important gap in the literature, this study used contingency theory and investigated the association between a variety of contextual (contingent and institutional) factors and the diffusion of MAIs in Libyan manufacturing and non-manufacturing organisations. Seven MAIs were chosen from the literature perceived to have higher popularity, namely, ABC, ABM, BSC, TC, life-cycle costing, benchmarking, and Kaizen. A questionnaire acted as the data collection instrument. Two hundred and fifty questionnaires were distributed, and one hundred and three useable questionnaires were returned. The results indicate that three factors were significantly associated with facilitating the adoption of MAIs in both sectors. They were using computer systems for MA purposes, top management support, and MA training programmes.

**Keywords:** management accounting; innovations; contextual factors; manufacturing and non-manufacturing; Emerging Economies; Libya

## 1. Introduction

By adopting the diffusion of innovation theory, several studies have identified a vast majority of contextual factors as potential influencing factors which could affect the decisions to adopt these MAIs in organisations (Al-Omiri and Drury 2007; Tajeddini 2010; Tajeddini and Mueller 2009; Tajeddini and Trueman 2012; Yazdifar and Askarany 2012; Yazdifar et al. 2019). However, it is still unclear why the adoption of MAIs in less developed countries is low (Funnell and Williams 2014; Gallhofer and Haslam 2004; Hardy and Ballis 2005; Irvine 2005; Jacobs 2005; Jayasinghe and Soobaroyen 2009; Joannidès and Berland 2013; Juan Banos and Funnell 2015; McKernan and Kosmala 2007; Mutch 2016; Thoradeniya et al. 2015; Tinker 2004).

To address this important gap in the literature, this study used contingency theory and investigated the association between a variety of contextual (contingent and institutional) factors and the diffusion of MAIs in Libyan manufacturing and non-manufacturing organisations. Seven MAIs were chosen from the literature perceived to have higher popularity, namely, ABC, ABM, BSC, TC, life-cycle costing, benchmarking, and Kaizen. The main purpose of this study was to promote the understanding of the extent of diffusion of MAIs, and also to explore factors that influence the adoption of MAIs by Libyan organisations. This study focuses on both manufacturing and non-manufacturing companies as most of the previous studies focused on manufacturing companies, neglecting non-manufacturing firms, especially in less developed countries. The reason for choosing non-manufacturing companies is because this sector has not been covered in depth in previous studies in Libya.

The researchers desired to cover different activities in the services sector such as telecom companies, financial organisations, hotels, and hospitals to explore their adoption of MAIs in Libya.

This study adopted a rational interpretation approach to explore key factors that facilitate the adoption of MAIs in Libya as well as examining the extent of their adoption. It further applied a combination of contingency and New Institutional Sociology (NIS) theories as its theoretical framework. Adopting both theories will help to focus on contingent and external organisational factors, including potential cultural factors which may influence the MAI adoption process. This study examines both contingent and institutional factors in the adoption and diffusion of MAIs and answers the following questions:

- Q1—Are there significant differences between adopting MAIs in the manufacturing and non-manufacturing sectors?
- Q2—What are the factors influencing the adoption of MAIs in the Libyan manufacturing and non-manufacturing sectors?

There are several gaps in the literature that the current study aims to overcome as follows:

- There is a rarity of studies that investigate the factors that influence the adoption of MAIs in Libya.
- None of the previous studies discussed, in detail, the status of MAIs in both manufacturing and non-manufacturing sectors.
- The previous studies either used contingency theory (for instance, Alkizza 2006; Abugalia 2011) or institutional theory (e.g., Alhashmi 2014; Leftesi 2008; Zoubi 2011), and no study used a combination of contingency theory and institutional theory as a theoretical framework.
- No previous study focused on a group of MAIs at the same time in Libya. This study tests seven chosen MAIs as a group and shows the impact of the independent factors on every chosen MAI separately.
- Finally, due to the limitation of several variables as well as the selected case studies in the past surveys, some scholars (e.g., Alhashmi 2014; Leftesi 2008; Zoubi 2011) proposed that future studies have an opportunity to examine the impact of missing variables and adopt a larger-scale survey approach to generalise the results to other settings statistically.

This paper contains six sections. The next section presents the relevant literature review which includes management accounting innovations and the investigation stage, contextual factors addressed in this study, and the adopted framework. Section 3 presents the methodology of this study. Section 4 summarises the main study findings and the limitations of this study and recommends future study topics. Section 5 discusses the result of this study, while Section 6 is devoted to the conclusion and contributions.

## 2. Literature Review

### 2.1. Management Accounting Innovations and the Investigation Stage

The extant literature has identified several responses to establish the cause of changes in the adoption of MAIs. Innes and Mitchell (1990) argued that the adoption of MAIs is a result of various contingency factors such as a competitive and dynamic market, product cost structures, management influence, and deteriorating financial performance. Furthermore, Scapens (2006) contended that business environment changes, including globalisation, customer focus, technological changes, and the changing face of organisational structures, have impacted the management information needs and therefore MAPs. In similar research by Yazdifar and Tsamenyi (2005), they argued that the adoption of MAIs was due to associated improvements in information technology, management style changes, a customer-oriented focus, restructuring of organisations, and globalisation.

In the same context, some studies were carried out in less developed countries such as the study by Nassar et al. (2011), who conducted a study aiming to assess the role of supply

factors in implementing (or not) MAIs among the Jordanian manufacturing sector. The study focused on seven factors including consultant companies; accounting education in Jordanian schools and universities; professional accounting bodies in Jordan; conferences, seminars, and workshops; co-operation between universities (academics) and companies (professionals); specialist MA journals; and accounting research in Jordan. They found that the most important factors leading to the decision to implement MAIs in the Jordanian manufacturing sector from a supply side perspective were consultant companies and accounting education. Moreover, the lack of co-operation between universities (academics) and companies (professionals) in Jordan, the lack of conferences, seminars, and workshops in Jordan, and the lack of local consultant companies were the main factors behind not adopting and implementing MAIs.

A study was undertaken in South Africa by Waweru et al. (2004) covering four retail firms to understand the process of MA change in these firms. The study suggested that the two major contingent factors influencing MA change were the intensified global competition and changes in technology. Additionally, the shortage of resources required to fund change, change resistance within employees, and fear of change were the dominant factors that impeded MA change.

Joshi (2001) examined the MAPs in use in India by surveying 60 large- and medium-sized industrial firms in India. The study found that the main factor influencing the adoption of MAIs was the size of organisations. Additionally, the conservative attitude of Indian management, autocratic leadership, and long-term orientation were other factors that influenced the adoption of MAIs. Wu et al. (2007) found that the type of ownership plays a role in structuring MAS in China when they studied both state-owned companies and joint ventures with foreign companies. The results indicated that joint ventures with foreign companies used MAIs more than local state-owned companies. In a different study, Joshi et al. (2011) examined how MAPs diffuse and are adopted among listed firms in the Gulf Cooperation Council (GCC) countries. The study argued that the most influential organisational factors in MA change were power and politics. Additionally, Allahyari and Ramazani (2011) examined independent variables that impede MA change within different sized (small, middle, large, and very large) manufacturing firms in Iran, aiming to obtain a better understanding of the MA change process. The study examined seven factors, namely, lack of accounting employees, lack of competition resources, management stability, problems in management, lack of accounting power, being assured of meeting legal requirements, and the lack of independence from the parent company. The results indicated that the lack of accounting employees, lack of independence from the parent company, and the lack of competition resources have a significant influence on MA change. Oyewo (2021) carried out a study in Nigeria, and the results showed that the overall usage rate of strategic management accounting (SMA) as innovation is moderate. The study found that environmental uncertainty causes a significant difference in the intensity of SMA usage across industries in the manufacturing sector.

In the Libyan context, some research has been conducted related to MA in Libya such as that by (Abugalia 2011; Abulghasim 2006; Alkizza 2006; Leftesi 2008; Zoubi 2011). Abulghasim (2006) studied MAPs used in Libyan state-owned firms. He found that the most significant factors that impeded the diffusion of MAPs were: a shortage of modern textbooks and publications, MA education, lack of training programmes, lack of competent operations managers, lack of an active professional MA society, lack of existing foreign companies, social, political, and cultural obstacles, and lack of financial resources. Additionally, other factors were less influential on the diffusion of MAPs such as the lack of MA studies, the lack of top management support, and lack of English language speakers. Similarly, Alkizza (2006) conducted a study to explore the MAPs in use in the Libyan context. He adopted Innes and Mitchell (1990) framework in his study. The study reported that the use of MAPs in Libya was motivated by four factors: change in the state regulations, change in the firm's strategic goals, increase in the market competition, and change in the organisational structure. The catalysts of change were the loss of market

share and poor financial performance, while the availability of academically qualified accountants who have limited ability in developing accounting systems, the availability of adequate computing resources, the autonomy of management from the parent company before becoming a unitary firm, the authorisation of accountants to change and improve the internal accounting methods, and the help of external accounting and computing advisors were the facilitators.

As we can see, none of the previous studies examined both manufacturing and non-manufacturing sectors in one study, in addition to the limited number of factors employed in those studies. This study was carried out to cover these limitations.

### 2.2. Contextual Factors Addressed in This Study

Innes and Mitchell (1990) argued that MA change does not occur as a result of one individual originating factor. Rather, it happens due to an association of a range of factors with each specific development. Therefore, the change in MA is a sophisticated process comprising a contribution of various factors. These different factors can be classified into two main categories, as follows:

- Macro-context factors (institutional/external factors);
- Micro-organisational factors (contingent/internal factors).

After reviewing the relevant literature and similar studies conducted in the same area, 21 factors were chosen to assess the most influential factors that may facilitate adopting MAIs in Libya. These factors are (8) institutional factors and (13) contingent factors.

#### 2.2.1. Macro-Context Factors (Institutional/External Factors)

The external environment of the business where a company operates might be certain or uncertain, plain or compound, or stable or moving (Fisher 1995). However, studying the external environment mainly represents looking at the uncertainty level. Uncertainty is described as the lack of availability of information required to make suitable decisions. Thus, to improve the decision making process, more detailed information is needed to eliminate environmental uncertainty. Many macro factors drive the change process, such as economic pressures, coercive pressures, normative pressure, and mimetic pressure (Granlund and Lukka 1998).

These factors are consistent with the study's framework, which adopted the institutional theory to understand and explain the macro/external factors that may cause MA change. The following sub-sections explain all these factors in more detail.

1.  Economic pressures: according to Granlund and Lukka (1998, p. 157), economic pressures comprise many different economic factors such as global economic fluctuations, recessions, and deregulation of markets; increased competition; advanced production technology (e.g., JIT); and advanced manufacturing technology (e.g., integrated systems such as SAP and the Internet).
2.  Coercive Pressures: These include two groups of factors. The first group represents factors driving conversions such as transnational legislation (e.g., European Union); transnational trade agreements (e.g., GATT/WTO, NAFTA, and EU); harmonisation of the financial accounting legislation; and headquarters' influence in general. The second group represents factors driving divergence such as national legislation, and national institutions/regulations (labour unions and financial institutions) (Granlund and Lukka 1998).
3.  Normative pressures: Two normative factors may drive convergence: management accountants' professionalisation, and university research and teaching, while national cultures and corporate cultures are considered as divergence driving factors.
4.  Mimetic processes: according to Granlund and Lukka (1998), memetic factors driving convergence are imitation of the leading company's practice and the international/global consultancy industry.

In this study, eight institutional factors were chosen as they were considered to have a significant influence on adopting MAIs in Libya, namely:

- Conferences, seminars, consultations, and workshops;
- Co-operation between universities (academic) and companies (professionals);
- Accounting research in Libya;
- Accounting education in Libya;
- Management accounting training in Libya;
- Professional accounting bodies in Libya;
- Headquarters and governmental regulations;
- Specialist management accounting journals.

2.2.2. Micro-Organisational Factors (Contingent/Internal Factors)

Micro-organisational factors refer to factors that exist inside the organisation. These factors comprise the organisational structure, managerial policies, production technology, employees, problems of existing techniques, and deterioration of financial performance (Alhashmi 2014). In terms of organisational structure, Abdel-Kader and Luther (2006) argued that this factor is considered one of the most significant factors that influence MAPs. Similarly, Haldma and Lääts (2002) found that there is evidence that the change in MAPs was linked with alteration in organisational characteristics such as the organisational structure. In their study, Innes and Mitchell (1990) focused on the role of organisational structure in the process of MA system change. Organisations that do not have a suitable allocation system of overheads will face a problem caused by the high rate of technology change in the production process. Where deterioration of financial performance occurs due to defective financial performance, it produces warnings to the organisation management to take the right steps that will improve performance and productivity. These steps are part of the change process when the organisation adopts new MA systems aiming to avoid any future financial deterioration. Thus, the new adopted MAPs and methods are needed to achieve a change decision by the top management. Innes and Mitchell (1990) described the deterioration of financial performance as a catalyst that pushes the adoption of new MAPs and methods in the unstable world of high-technology organisations. Similarly, Haldma and Lääts (2002) reported that failure to receive the required information to help in making decisions could be considered as an important catalyst in developing the cost and MA system. On the other hand, Ax and Greve (2017) carried out a study to test the effect of a firm's values and beliefs on the diffusion of innovation among Swedish manufacturing firms. They assumed that a diffusing innovation that is compatible with a firm's values and beliefs is adopted early if it is perceived as delivering adequate gains, while the innovation is rejected if it is not perceived as doing so. They found that in most respects, the results support their assumptions.

The contingent factors employed in this study were derived mainly from Innes and Mitchell (1990) study, and the literature (studies such as Chenhall et al. 1981; Cobb et al. 1995; Haldma and Lääts 2002; Merchant 1981). This study includes thirteen contingent factors (see Table 1) that serve the objective of this study as follows:

**Table 1.** Contingent factors.

| Variable (Factor) | The Source That Factors Were Taken From |
|---|---|
| Company structure (centralisation and decentralisation) | Innes and Mitchell (1990)/Merchant (1981) |
| Company size | Haldma and Lääts (2002)/Merchant (1981)/Chenhall et al. (1981) |
| The availability of adequate accounting staff | Innes and Mitchell (1990)/Haldma and Lääts (2002)/Cobb et al. (1995) |
| Using a computer system for MA purposes | Innes and Mitchell (1990) |
| The authority attributed to the accounting function | Innes and Mitchell (1990) |
| The competitiveness of the market | Innes and Mitchell (1990)/Haldma and Lääts (2002)/Cobb et al. (1995) |
| Production technology | Innes and Mitchell (1990)/Haldma and Lääts (2002) |
| Product cost structure | Innes and Mitchell (1990)/Haldma and Lääts (2002) |
| The loss of market share | Innes and Mitchell (1990) |
| The arrival of new accountants | Innes and Mitchell (1990) |
| Deterioration in profitability | Innes and Mitchell (1990) |
| Top management support | Cobb et al. (1995) |
| Adequate financial resources | Haldma and Lääts (2002) |

*2.3. The Adopted Framework*

This study's framework model represents a combination of contingency and institutional theories (see Figure 1). The framework shows the factors that influence the adoption of MAIs among Libyan manufacturing and non-manufacturing organisations. The institutional theory side of this model is based mainly on DiMaggio and Powell (1983) study, where they divided the institutional factors into three types of isomorphism, namely, coercive, mimetic, and normative. On the other hand, the contingency side of this model is based mainly on the model developed by Innes and Mitchell (1990) and Cobb et al. (1995), which explains the process of MA change that comprises motivators, catalysts, facilitators, and other contingent factors added by Cobb et al.'s (1995) study.

In this study's framework, the institutional and contingency factors that cause MA changes are opposed by barriers that prevent or impede the change process. All contingency and institutional factors that pass the barriers to change may interact together, causing MA change in the organisation and the adoption of MAIs is an outcome of this process. The following lines explain the reasons to combine the two theories.

Ketokivi and Schroeder (2004) examined the implementation of specific practices using contingency and institutional perspectives; they found that the contingency approach failed to provide a comprehensive explanation to why certain organisations adopted certain practices. On the other hand, the mimicry argument provides an adequate explanation of the phenomenon. In a different study, Williams and Seaman (2001) found that several variables not included in the contingency theory may impact MA change. Accordingly, they contended that the contingency theory provides a limited explanation of the process of MA change. Moreover, they stated that "additional variables could be added to the model to refine measurements" (p. 457).

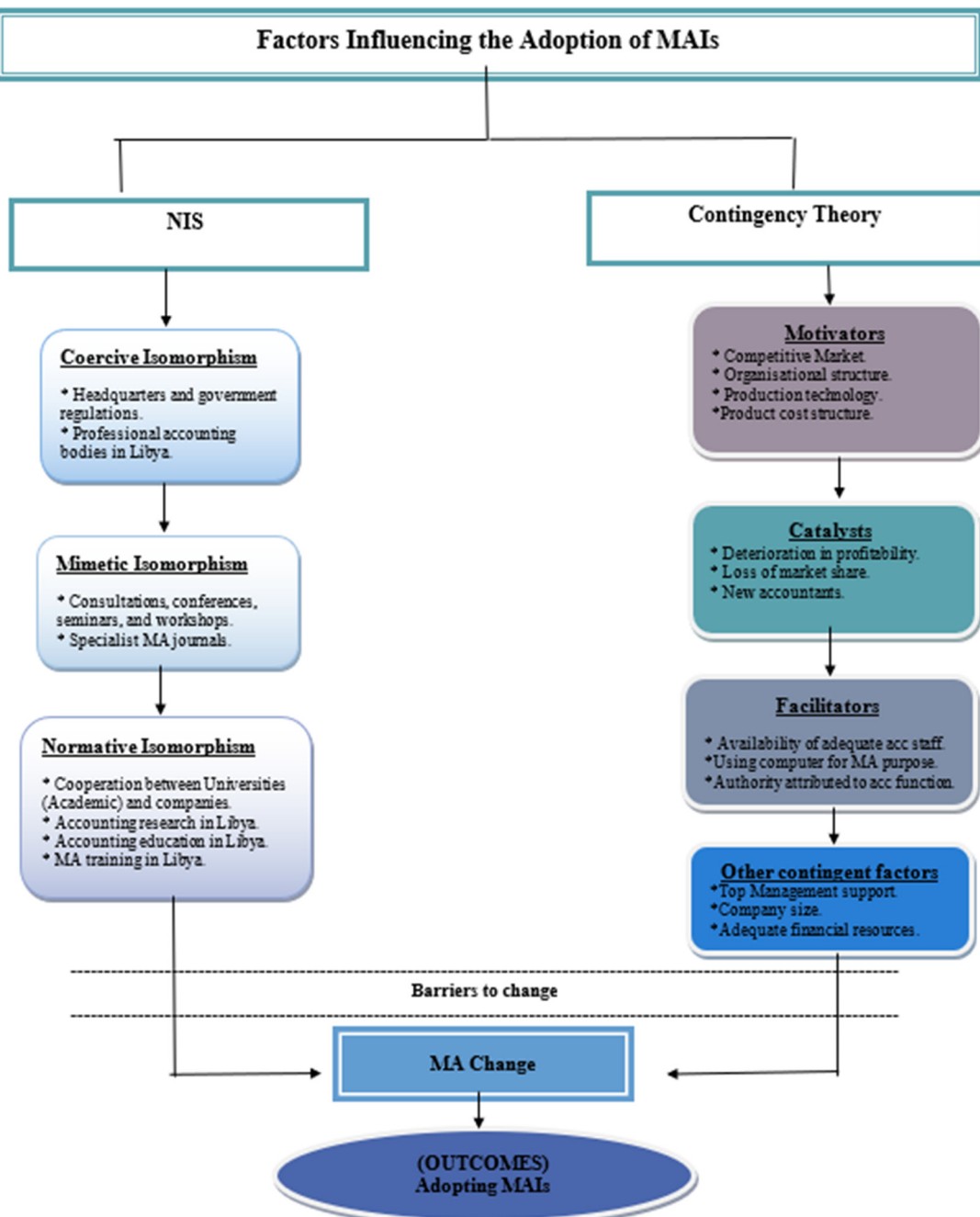

**Figure 1.** The study's theoretical framework.

Regarding the institutional theory, Yazdifar (2004) reported that institutional theory focuses on the macro impact of the external environment. Therefore, it is unable to provide a holistic explanation of the process of MA change. To overcome the limitations of both the contingency theory and the NIS theory, we adopted both. Contingency fit and institutional fit provide complementary and interdependent explanations of a firm's performance (Volberda et al. 2012).

Reviewing the literature shows that many studies discussed the combination and the integration of contingency and institutional theory. Volberda et al. (2012) conducted a study to test their framework using data collected from 3259 respondents in 1904 companies; the result indicated that "contingency and institutional fit are complementary and interrelated explanations of firm performance and show that the combination of both theories

produces superior insights into the relationship between the fit and firm performance" (p. 1040). Carroll (1993) explained that a firm's successes are enhanced when there are complementarities between contingency and institutional theories for the understanding of the homogeneity or heterogeneity of firms in different industries. Similarly, Clark and Soulsby (1995) reported that the combination of contingency and institutional theories complemented each other and improved the insights gained related to organisational change among former enterprises in the Czech Republic.

According to Volberda et al. (2012), the integration between the contingency and institutional perspectives is vital because none of them can solely explain the success of the firm and its relationship with its environment. In the same context, Heugens and Lander (2009, p. 64) argued that "According to contingency theory, managers carefully analyse the firm's task environment, considering the internal characteristics of the firm, and adapt their practices accordingly. On the other hand, according to institutional theory, the environment exerts strong pressures for institutional fit or adoption of "conformance enhancing templates". In line with the presented literature review in this paper, the following hypotheses are proposed:

**Hypothesis 1 (H1).** *There are no significant differences between the adoption of MAIs in the manufacturing and non-manufacturing sectors.*

**Hypothesis 2 (H2).** *There are no significant differences among contextual factors (contingent and institutional) in terms of their level of associations with MAIs in the manufacturing and non-manufacturing sectors.*

### 3. Research Methodology

This research adopted two theories: contingency and NIS. The rationale behind using institutional theory with contingency theory is to overcome the disadvantages of each theory when used separately. Therefore, using hybrid theories can enhance our understanding of the MAP adoption process. In other words, using two different theories ensures more credibility in the research results through studying the influence of different external and internal factors on the MAI adoption process.

A survey questionnaire consist of three sections was distributed to collect quantitative data (for more details see Appendices A–C). This study assesses the status of adoption of MAIs and examines the factors influencing the adoption of MAIs in Libya in medium- and large-size manufacturing and non-manufacturing companies, whether they are private or state-owned companies. The reason for choosing large- and medium-size organisations is that size has an impact in terms of the adoption process of MAIs, and larger organisations are more likely to implement MAIs than smaller ones (Abdel-Kader and Luther 2006). The targeted population included those holding top financial positions, those who are responsible for MA activities in these institutions, or anyone else capable of filling in the questionnaire. Furthermore, it included individuals who are not in an executive position but connected with the study subject such as academic staff, and individuals who hold professional qualifications in accounting and particularly in MA.

The targeted sample in this study included manufacturing and non-manufacturing companies that operate in Libya. It is important to confirm the suitability of the sample size of this study. Pallant (2007) contended that a small sample size is when the number of cases is less than 50, whereas a large sample refers to a sample when the number of cases is more than 100. To generalise the findings of this study, the ratio of respondents to independent variables is important. In this context, Hair et al. (2009) contended that for each independent variable, there must be four respondents as a minimum acceptable ratio, and the desirable ratio is between 10 and 20 respondents for each independent variable. The ratio in this study was about 5:1 (103/21), and thus the level of the ratio of the respondents (cases) to independent variables is acceptable.

To guarantee at least four respondents to each independent variable and to collect a minimum of 100 useable questionnaire forms, a 40% response rate was set as a target.

Accordingly, 250 questionnaires were distributed to cover 80 manufacturing organisations and 170 non-manufacturing ones (based on their total populations). The sample of this study covered organisations based in the greater Tripoli area as this area was considered relatively safer than other parts of Libya. Table 2 presents a summary of the response rates.

**Table 2.** Response rates.

| Description | Sample | | Total | % |
|---|---|---|---|---|
| | **Manufacturing** | **Non-Manufacturing** | | |
| **Distributed questionnaires** | 80 | 170 | 250 | 100 |
| **Total number of returned questionnaires** | 49 | 72 | 121 | 48.4 |
| **(-) Incomplete questionnaires** | 5 | 13 | (18) | (7.2) |
| **Total number of useable questionnaires** | 44 | 59 | 103 | 41.2 |

Data screening, descriptive statistics related to the factors that may influence the adoption of MAIs, and assessment of the status of MAIs in the Libyan manufacturing and non-manufacturing organisations were conducted. Descriptive statistics are useful to show the most influential factors on the adoption of MAIs, and also to present the adoption rate of MAIs in the Libyan manufacturing and non-manufacturing organisations in separate tables.

An independent sample *t*-test was carried out to compare the differences in the MAI means between the Libyan manufacturing and non-manufacturing organisations.

## 4. Results and Main Findings

The contents of the 103 valid questionnaires were entered into SPSS software to produce introductory descriptive statistics which were helpful in terms of assuring accuracy, and testing for missing data values, normality, and outliers.

### 4.1. Reliability and Validity

The data obtained from the questionnaire were analysed using SPSS software. Hair et al. (2009) stated that measuring reliability provides a good indication of the consistency degree.

In this study, Cronbach's alpha value for MAIs that contains seven items is 0.859, which can be classified as good (see Table 3).

**Table 3.** Reliability statistics of MAIs.

| Cronbach's Alpha | Number of Items |
|---|---|
| 0.859 | 7 |

Regarding factors that may facilitate adoption of MAIs decisions, this study included 21 variables. The reliability statistics were tested, showing that the value of Cronbach's alpha is 0.834, while the test of the reliability for statistics of factors that may impede the adoption of MAIs, including 24 variables, shows that the value of Cronbach's alpha is 0.838.

Regarding finding validity, Tashakkori and Teddlie (1998) suggested that there are two main types of validity in quantitative research. The first type is external validity, which refers to the ability of the researcher to generalise the research results to other cases or other groups of people. The second type of validity is internal validity, which refers to the extent of confidence about the findings of the research about the relationship between variables and the degree of systemic error.

To achieve external validity, the sample was chosen to represent the study population in terms of the variety of sectors, ownership, and the organisation's size.

To ease the data entry process and to detect any missed values, all valid physical questionnaires were given the same number of electronic cases in the SPSS software to help to compare electronic cases with the original (printed questionnaires) sources of the data. This process facilitated the re-entry of any missed data values when they were detected. Regarding outliers and normality, the data related to the status of MAIs in Libya and the factors that influence the adoption of MAIs were tested. According to Hair et al. (2009), the value ±1.96 is considered a cut-off point for both kurtosis and skewness. This test was conducted to explore the ratios of skewness and kurtosis, which were in the acceptable range, as it is shown in Table 4.

**Table 4.** Descriptive statistics for factors facilitating the adoption of MAIs.

| Factor | N | Mean | Std. Deviation | Skewness | | Kurtosis | |
|---|---|---|---|---|---|---|---|
| | Statistic | Statistic | Statistic | Statistic | Std. Error | Statistic | Std. Error |
| Business dependency | 103 | 1.46 | 0.501 | 0.178 | 0.238 | −2.008 | 0.472 |
| Number of employees | 103 | 3.85 | 1.132 | −0.410 | 0.238 | −1.285 | 0.472 |
| The availability of adequate accounting staff | 103 | 3.95 | 0.974 | −0.746 | 0.238 | −0.060 | 0.472 |
| Using computer systems for MA purposes | 103 | 4.41 | 0.747 | −1.118 | 0.238 | 0.734 | 0.472 |
| The authority attributed to the accounting function | 103 | 3.88 | 0.820 | −0.432 | 0.238 | −0.211 | 0.472 |
| The competitiveness of the market | 103 | 3.70 | 1.074 | −0.581 | 0.238 | −0.375 | 0.472 |
| Production technology | 103 | 3.57 | 1.151 | −0.712 | 0.238 | −0.269 | 0.472 |
| The loss of market share | 103 | 2.71 | 1.265 | 0.036 | 0.238 | −1.140 | 0.472 |
| Competent accountants | 103 | 3.95 | 1.033 | −0.937 | 0.238 | 0.453 | 0.472 |
| Deterioration in profitability | 103 | 2.92 | 1.234 | −0.105 | 0.238 | −0.938 | 0.472 |
| Top management support | 103 | 4.23 | 0.843 | −1.269 | 0.238 | 1.962 | 0.472 |
| Conferences, seminars, and workshops | 103 | 3.47 | 1.127 | −0.081 | 0.238 | −0.928 | 0.472 |
| Co-operation between universities (academics) and companies (professionals) | 103 | 3.38 | 1.246 | −0.416 | 0.238 | −0.807 | 0.472 |
| Accounting research in Libya | 103 | 3.47 | 1.119 | −0.192 | 0.238 | −0.906 | 0.472 |
| Accounting education in Libya | 103 | 3.84 | 1.118 | −0.845 | 0.238 | −0.043 | 0.472 |
| Management accounting training programmes | 103 | 4.07 | 1.012 | −1.121 | 0.238 | 0.729 | 0.472 |
| Adequate financial resources | 103 | 3.75 | 1.055 | −0.600 | 0.238 | −0.223 | 0.472 |
| Professional accounting bodies in Libya | 103 | 2.96 | 1.335 | 0.148 | 0.238 | −1.265 | 0.472 |
| Product cost structure | 103 | 2.80 | 1.240 | 0.146 | 0.238 | −1.080 | 0.472 |
| Headquarters and government regulation | 103 | 3.6990 | 0.88379 | −0.496 | 0.238 | −0.370 | 0.472 |
| Specialist management accounting journals | 103 | 3.12 | 1.331 | −0.065 | 0.238 | −1.238 | 0.472 |
| Valid N (list-wise) | 103 | | | | | | |

As shown in Table 5, the non-manufacturing sector comprises the highest percentage of respondents (57.2%), and it includes the following:

- Finance (including banking and insurance) 15.5%;
- Information technology (including telecom, telephone, and Internet) 11.6%;
- Transport (including road, sea, and air transport) 2.9%;
- Commerce (including retail, wholesale, import, and export trading) 6.8%;
- Hotels 3.9%;
- Health services 8.7%;

- Construction 7.8%.

**Table 5.** Types of businesses in this study.

| | Manufacturing | | Non-Manufacturing | | Cumulative Percent |
|---|---|---|---|---|---|
| | Frequency | % | Frequency | % | |
| Manufacturing | 29 | 28.2 | | | 28.2 |
| Oil and Gas | 15 | 14.6 | | | 42.8 |
| Construction | | | 8 | 7.8 | 50.6 |
| Finance (including banking and insurance) | | | 16 | 15.5 | 66.1 |
| Information technology (including telecom, telephone, and Internet) | | | 12 | 11.6 | 77.7 |
| Transport (including road, sea, and air transport) | | | 3 | 2.9 | 80.6 |
| Commerce (including retail, wholesale, import, and export trading) | | | 7 | 6.8 | 87.4 |
| Commerce (including retail, wholesale, import, and export trading) | | | 4 | 3.9 | 91.3 |
| Health services | | | 9 | 8.7 | 100 |
| Total | 44 | 42.8 | 59 | 57.2 | 103 |

The manufacturing sector represents 42.8% of all the respondents.

### 4.2. The Status of the Adoption of MAIs in Libya

This section examines the extent of the use of "MAIs" in the Libyan manufacturing and non-manufacturing sectors.

The respondents were asked to answer the questions by ticking one of the listed statements which best described the status of MAIs in their organisations. A five-point Likert scale was used: 1 (never heard of it), 2 (never considered adoption), 3 (considered then rejected), 4 (under consideration), and, finally, 5 (adopted and currently in use). Seven MAIs were chosen from the relevant literature and previous studies after considering what might suit less developed countries such as Libya. The result indicates that the adoption rate of MAIs, in general, is lower than traditional MAPs. Furthermore, the adoption is still in its infancy compared to developed countries.

Based on the "mean" value shown in Table 6, the MAIs were sorted in order from highest to lowest adoption rates. The ABC technique came first, adopted by 43.2% of the respondents. Benchmarking costing came second, with an adoption rate of 27.3%, followed by Kaizen costing in third place, with a 27.3% adoption rate. Target costing came fourth with 20.5%. Activity-based management, life-cycle costing, and balanced scorecard occupied the fifth to seventh places, with adoption rates of 11.4%, 11.4%, and 2.3%, respectively.

**Table 6.** Status of MAIs in the manufacturing sector in Libya.

| Technique | N | Adoption Rate % | Mean | Std. Deviation |
|---|---|---|---|---|
| Activity-Based Costing | 44 | 43.2 | 4.18 | 1.88 |
| Benchmarking | 44 | 27.3 | 3.0 | 2.1 |
| Kaizen Costing | 44 | 27.3 | 2.95 | 1.99 |
| Target Costing | 44 | 20.5 | 3.02 | 1.78 |
| Activity-Based Management | 44 | 11.4 | 2.52 | 1.62 |
| Life-Cycle Costing | 44 | 11.4 | 2.43 | 1.5 |
| Balanced Scorecard | 44 | 2.3 | 2.45 | 1.34 |

### 4.3. Contextual Factors and the Adoption of MAIs in the Manufacturing Sector

In this part, the respondents were asked to specify the importance of each factor in terms of facilitating the adoption process by choosing one of the values on a five-point Likert scale sorted in order from 1 = do not facilitate to 5 = extremely facilitate.

Table 7 shows the details of 44 useable questionnaires collected from manufacturing organisations. The 21 factors' importance and ranking according to their mean are shown in Table 7. However, Table 8 represents factors that belong to contingency theory, while Table 9 shows institutional factors.

**Table 7.** Contextual factors for the adoption of MAIs in the manufacturing sector.

| Factor | N | Sum | Mean | Rank |
|---|---|---|---|---|
| **Using computer systems for MA purposes** | 44 | 194 | 4.41 | 1 |
| **Top management support** | 44 | 193 | 4.39 | 2 |
| **Management accounting training programmes** | 44 | 189 | 4.30 | 3 |
| **Accounting education in Libya** | 44 | 182 | 4.14 | 4 |
| **The arrival of new accountants** | 44 | 177 | 4.02 | 5 |
| **Production technology** | 44 | 171 | 3.89 | 6 |
| **The authority attributed to the accounting function** | 44 | 170 | 3.86 | 7 |
| **The availability of adequate accounting staff** | 44 | 169 | 3.84 | 8 |
| **The competitiveness of the market** | 44 | 166 | 3.77 | 9 |
| **Headquarters and government regulation** | 44 | 162 | 3.68 | 10 |
| **Accounting research in Libya** | 44 | 161 | 3.66 | 11 |
| **Adequate financial resources** | 44 | 161 | 3.66 | 12 |
| **Co-operation between universities (academics) and companies (professionals)** | 44 | 158 | 3.59 | 13 |
| **Conferences, seminars, and workshops** | 44 | 156 | 3.55 | 14 |
| **Company size** | 44 | 150 | 3.41 | 15 |
| **Deterioration in profitability** | 44 | 132 | 3.00 | 16 |
| **Professional accounting bodies in Libya** | 44 | 127 | 2.89 | 17 |
| **Specialist management accounting journals** | 44 | 126 | 2.86 | 18 |
| **The loss of market share** | 44 | 122 | 2.77 | 19 |
| **Product cost structure** | 44 | 120 | 2.73 | 20 |
| **Company structure** | 44 | 118 | 2.66 | 21 |

**Table 8.** Contingency factors facilitating the adoption of MAIs in the manufacturing sector.

| Factor | Sum | Mean | Rank |
|---|---|---|---|
| **Using computer systems for MA purposes** | 194 | 4.41 | 1 |
| **Top management support** | 193 | 4.39 | 2 |
| **The arrival of new accountants** | 177 | 4.02 | 3 |
| **Production technology** | 171 | 3.89 | 4 |
| **The authority attributed to the accounting function within the organisation** | 170 | 3.86 | 5 |
| **The availability of adequate accounting staff** | 169 | 3.84 | 6 |
| **The competitiveness of the market** | 166 | 3.77 | 7 |
| **Adequate financial resources** | 161 | 3.66 | 8 |
| **Company size** | 150 | 3.41 | 9 |
| **Deterioration in profitability** | 132 | 3.00 | 10 |
| **The loss of market share** | 122 | 2.77 | 11 |
| **Product cost structure** | 120 | 2.73 | 12 |
| **Company structure (centralisation and decentralisation)** | 118 | 2.68 | 13 |

**Table 9.** Institutional factors facilitating the adoption of MAIs in the manufacturing sector.

| Factor | Sum | Mean | Rank |
|---|---|---|---|
| **Management accounting training programmes** | 189 | 4.30 | 1 |
| **Accounting education in Libya** | 182 | 4.14 | 2 |
| **Headquarters and governmental regulations** | 162 | 3.68 | 3 |
| **Accounting research in Libya** | 161 | 3.66 | 4 |
| **Co-operation between universities (academics) and companies (professionals)** | 158 | 3.59 | 5 |
| **Conferences, seminars, and workshops** | 156 | 3.55 | 6 |
| **Professional accounting bodies in Libya** | 127 | 2.89 | 7 |
| **Specialist management accounting journals** | 126 | 2.86 | 8 |

To assess the influence of the factors that belong to contingency and institutional theories on the adoption of MAIs within Libyan organisations, the factors were divided into two groups, as shown in Table 8 (which contains contingent factors) and Table 9 (which contains institutional factors). It can be seen from Table 8 that there are ten factors related to contingency theory, which have a significant influence on adopting MAIs, with mean values ranging between 3.00 and 4.41. Moreover, seven of these factors are among the top ten factors that have the most influence on the adoption of the MAI process.

Regarding factors related to institutional theory, Table 9 shows the mean values and ranks of these factors. The mean value in this group ranges between 4.30 and 2.86, which is lower than the mean value of the contingency group. Moreover, there are just three institutional factors among the top ten factors that have the most influence on adopting MAIs within Libyan manufacturing organisations, and they are ranked 3, 4 and 10, while the other five factors ranked between 11 and 18.

*4.4. The Status of the Adoption of MAIs in the Non-Manufacturing Sector*

Table 10 shows the status of adoption of MAIs in the non-manufacturing sector. The result indicates that the adoption rate of MAIs is still in its infancy compared to developed countries. Based on the "mean" value shown in Table 10, the MAIs were sorted in order from highest to lowest adoption rates. The ABC technique came first, adopted by 20.3% of respondents' organisations. Benchmarking came second, with an adoption rate of 18.6%, followed by Kaizen costing in third place, with 16.9% adoption. Life-cycle costing came fourth, with an 11.9% adoption rate, and target costing, activity-based management, and balanced scorecard occupied the fifth to seventh places, with adoption rates of 10.2%, 10.2%, and 5.1%, respectively.

**Table 10.** Status of MAIs in the non-manufacturing sector in Libya.

| Technique | N | Adoption Rate % | Mean | Std. Deviation |
|---|---|---|---|---|
| **Activity-Based Costing** | 59 | 20.3 | 3.32 | 1.77 |
| **Benchmarking** | 59 | 18.6 | 2.69 | 1.93 |
| **Kaizen Costing** | 59 | 16.9 | 2.76 | 1.88 |
| **Life-Cycle Costing** | 59 | 11.9 | 2.54 | 1.67 |
| **Target Costing** | 59 | 10.2 | 2.61 | 1.59 |
| **Activity-Based Management** | 59 | 10.2 | 2.37 | 1.63 |
| **Balanced Scorecard** | 59 | 5.1 | 2.47 | 1.55 |

*4.5. Contextual Factors and the Adoption of MAIs in the Non-Manufacturing Sector*

Table 11 shows the details of 59 useable questionnaires received from the non-manufacturing sector regarding the different factors' importance and ranking according to their mean. Tables 12 and 13 present factors that belong to both contingency and institutional factors.

**Table 11.** Contextual factors and the adoption of MAIs in the non-manufacturing sector.

| Factor | N | Sum | Mean | Rank |
|---|---|---|---|---|
| Using computer systems for MA purposes | 59 | 260 | 4.41 | 1 |
| Top management support | 59 | 243 | 4.12 | 2 |
| The availability of adequate accounting staff | 59 | 238 | 4.03 | 3 |
| Management accounting training programmes | 59 | 230 | 3.90 | 4 |
| The arrival of new accountants | 59 | 230 | 3.90 | 5 |
| The authority attributed to the accounting function | 59 | 230 | 3.90 | 6 |
| Adequate financial resources | 59 | 225 | 3.81 | 7 |
| Headquarters and government regulation | 59 | 219 | 3.71 | 8 |
| The competitiveness of the market | 59 | 215 | 3.64 | 9 |
| Accounting education in Libya | 59 | 214 | 3.63 | 10 |
| Company size | 59 | 213 | 3.61 | 11 |
| Conferences, seminars, and workshops | 59 | 201 | 3.41 | 12 |
| Production technology | 59 | 197 | 3.34 | 13 |
| Accounting research in Libya | 59 | 196 | 3.32 | 14 |
| Specialist management accounting journals | 59 | 195 | 3.31 | 15 |
| Co-operation between universities (academics) and companies (professionals) | 59 | 190 | 3.22 | 16 |
| Professional accounting bodies in Libya | 59 | 178 | 3.02 | 17 |
| Deterioration in profitability | 59 | 169 | 2.86 | 18 |
| Product cost structure | 59 | 168 | 2.85 | 19 |
| The loss of market share | 59 | 157 | 2.66 | 20 |
| Company structure (centralisation and decentralisation) | 59 | 153 | 2.59 | 21 |

**Table 12.** Contingency factors facilitating the adoption of MAIs in the non-manufacturing sector.

| Factor | Sum | Mean | Rank |
|---|---|---|---|
| Using computer systems for MA purposes | 260 | 4.41 | 1 |
| Top management support | 243 | 4.12 | 2 |
| The availability of adequate accounting staff | 238 | 4.03 | 3 |
| The arrival of new accountants | 230 | 3.9 | 4 |
| The authority attributed to the accounting function within the organisation | 230 | 3.9 | 5 |
| Adequate financial resources | 225 | 3.81 | 6 |
| The competitiveness of the market | 215 | 3.64 | 7 |
| Company size | 213 | 3.61 | 8 |
| Production technology | 197 | 3.34 | 9 |
| Deterioration in profitability | 169 | 2.86 | 10 |
| Product cost structure | 168 | 2.85 | 11 |
| The loss of market share | 157 | 2.66 | 12 |
| Company structure (centralisation and decentralisation) | 153 | 2.59 | 13 |

**Table 13.** Institutional factors facilitating the adoption of MAIs in the non-manufacturing sector.

| Factor | Sum | Mean | Rank |
|---|---|---|---|
| Management accounting training programmes | 230 | 3.9 | 1 |
| Headquarters and governmental regulations | 219 | 3.71 | 2 |
| Accounting education in Libya | 214 | 3.63 | 3 |
| Conferences, seminars, and workshops | 201 | 3.41 | 4 |
| Accounting research in Libya | 196 | 3.32 | 5 |
| Specialist management accounting journals | 195 | 3.31 | 6 |
| Co-operation between universities (academics) and companies (professionals) | 190 | 3.22 | 7 |
| Professional accounting bodies in Libya | 178 | 3.02 | 8 |

After reviewing the relevant literature and similar studies conducted in the same area, 21 factors were chosen to identify the most influential factors that may facilitate adopting

MAIs in Libya. These factors are contingent (13) factors and institutional (8) factors. The discussion related to Table 11 will be based on whether the nature of each factor belongs to contingency or institutional theory, in order to assess the influence of both theories on the adoption of MAIs.

To find out the influence of the contingency and institutional factors presented in Table 11 on the adoption of MAIs separately, we divided them into two different groups. Table 12 shows the contingency factors, and Table 13 includes the institutional factors.

Table 12 contains 13 contingent factors; a total of 7 out of the 13 factors are among the top ten that have the most influence on the adoption of MAIs in the non-manufacturing sector. Moreover, there are nine factors considered to have a significant impact on the adoption of MAIs, with mean values between 4.41 and 3.34.

The second group comprises institutional factors as they are shown in Table 13. This group consists of eight institutional factors which have mean values ranging between 3.90 and 3.02. Moreover, there are just three institutional factors among the top ten factors that have the most influence on the adoption of MAIs within Libyan manufacturing organisations ranked 4, 8, and 10, while the other five factors ranked between 12 and 17.

### 4.6. Hypothesis Testing

This section reports the results of the hypothesis testing. We used a test to check if the two means are significantly different from each other. Table 14 shows the basic information related to seven MAIs in terms of sample size (n), means, standard deviation, and standard error of the mean for both manufacturing and non-manufacturing sectors.

**Table 14.** Group statistics.

|  | Business Type | N | Mean | Std. Deviation | Std. Error Mean |
|---|---|---|---|---|---|
| Activity-Based Costing | Manufacturing | 44 | 4.18 | 1.883 | 0.284 |
|  | Non-Manufacturing | 59 | 3.32 | 1.766 | 0.230 |
| Activity-Based Management | Manufacturing | 44 | 2.52 | 1.621 | 0.244 |
|  | Non-Manufacturing | 59 | 2.37 | 1.628 | 0.212 |
| Balanced Scorecard | Manufacturing | 44 | 2.45 | 1.337 | 0.202 |
|  | Non-Manufacturing | 59 | 2.47 | 1.546 | 0.201 |
| Target Costing | Manufacturing | 44 | 3.02 | 1.798 | 0.271 |
|  | Non-Manufacturing | 59 | 2.61 | 1.587 | 0.207 |
| Life-Cycle Costing | Manufacturing | 44 | 2.43 | 1.500 | 0.226 |
|  | Non-Manufacturing | 59 | 2.54 | 1.675 | 0.218 |
| Benchmarking | Manufacturing | 44 | 3.00 | 2.091 | 0.315 |
|  | Non-Manufacturing | 59 | 2.69 | 1.932 | 0.252 |
| Kaizen Costing | Manufacturing | 44 | 2.95 | 1.988 | 0.300 |
|  | Non-Manufacturing | 59 | 2.76 | 1.879 | 0.245 |

A *t*-test is called an inferential statistic because it allows us to make inferences about the population beyond our data. A *t*-test has three different types:

1. Independent sample *t*-test, which tests the means of two different groups (e.g., the manufacturing versus the non-manufacturing sectors);
2. Paired sample *t*-test, which tests the mean of one group twice (test one group before and after an action or change);
3. One sample *t*-test, which tests the mean of one group against a set mean.

Accordingly, this study used an independent sample *t*-test as it is testing the difference between the adoption of MAIs in two different sectors/groups (manufacturing and non-manufacturing sectors). Table 15 shows the independent sample *t*-test for all seven MAIs

under study. Equal variances were assumed to provide results for the actual independent sample *t*-test, which included the following:

- *T* is the computed test statistic;
- df is the degrees of freedom;
- Sig. (2-tailed) is the *p*-value corresponding to the given test statistic and degrees of freedom;
- Mean Difference is the difference between the sample means; it also corresponds to the numerator of the test statistic;
- Std. Error Difference is the standard error; it also corresponds to the denominator of the test statistic.

**Table 15.** Independent sample test.

| | | *t*-Test for Equality of Means | | | | | | |
| | | *T* | df | Sig. (2-tailed) | Mean Difference | Std. Error Difference | 95% Confidence Interval of the Difference | |
| | | | | | | | Lower | Upper |
|---|---|---|---|---|---|---|---|---|
| Activity-Based Costing | Equal variances assumed | 2.376 | 101 | 0.019 | 0.860 | 0.362 | 0.142 | 1.578 |
| | Equal variances not assumed | 2.353 | 89.382 | 0.021 | 0.860 | 0.365 | 0.134 | 1.586 |
| Activity-Based Management | Equal variances assumed | 0.463 | 101 | 0.644 | 0.150 | 0.324 | −0.492 | 0.792 |
| | Equal variances not assumed | 0.463 | 93.020 | 0.644 | 0.150 | 0.324 | −0.493 | 0.792 |
| Balanced Scorecard | Equal variances assumed | −0.069 | 101 | 0.945 | −0.020 | 0.291 | −0.597 | 0.557 |
| | Equal variances not assumed | −0.070 | 98.731 | 0.944 | −0.020 | 0.285 | −0.585 | 0.545 |
| Target Costing | Equal variances assumed | 1.233 | 101 | 0.220 | 0.413 | 0.335 | −0.251 | 1.076 |
| | Equal variances not assumed | 1.211 | 85.965 | 0.229 | 0.413 | 0.341 | −0.265 | 1.090 |
| Life-Cycle Costing | Equal variances assumed | −0.346 | 101 | 0.730 | −0.111 | 0.319 | −0.744 | 0.523 |
| | Equal variances not assumed | −0.352 | 97.574 | 0.726 | −0.111 | 0.314 | −0.734 | 0.513 |
| Benchmarking | Equal variances assumed | 0.765 | 101 | 0.446 | 0.305 | 0.399 | −0.486 | 1.096 |
| | Equal variances not assumed | 0.757 | 88.570 | 0.451 | 0.305 | 0.403 | −0.496 | 1.106 |
| Kaizen Costing | Equal variances assumed | 0.500 | 101 | 0.618 | 0.192 | 0.384 | −0.569 | 0.953 |
| | Equal variances not assumed | 0.496 | 89.815 | 0.621 | 0.192 | 0.387 | −0.577 | 0.960 |

The results of testing the hypotheses of this study are as follows:

Hypothesis 1:

There are no significant differences between the adoption of MAIs in the manufacturing and non-manufacturing sectors.

The *t*-test was conducted to make sure if there is no significant difference between adopting ABC in the manufacturing and non-manufacturing sectors. From Table 15, we can see that *p*-value = 0.019, which is <0.05, meaning there is a statistically significant difference between the means of the two groups.

The *t*-test was carried out to find out if there is a significant difference between adopting ABM in the manufacturing and non-manufacturing sectors. Table 15 shows that *p*-value = 0.644, which is >0.05. This result means that there is no statistically significant difference between the means of the two samples.

From the *t*-test results shown in Table 15, *p*-value = 0.945, which is far greater than 0.05. This result shows a lack of statistically significant differences between the means of the two samples.

The *t*-test was undertaken to discover whether the mean difference in both samples is statistically significant. The result shows that *p*-value = 0.220, which is >0.05.

The *t*-test was carried out to find out if there is a significant difference between adopting life-cycle costing in the manufacturing and non-manufacturing sectors. Table 15 shows that *p*-value = 0.730, which is >0.05. This result means that there is no statistically significant difference between the means of the two samples.

The *t*-test was used to confirm if the difference in means between the manufacturing and non-manufacturing samples related to adopting benchmarking is statistically significant. Table 15 shows that *p*-value = 0.446, which is >0.05. This result means that there is no statistically significant difference between the means of the two samples.

The *t*-test was carried out to find out if there is a significant difference between adopting ABM in the manufacturing and non-manufacturing sectors. Table 15 shows that *p*-value = 0.618, which is >0.05. This result means that there is no statistically significant difference between the means of the two samples.

The descriptive analysis of the status of MAIs in the manufacturing and non-manufacturing sectors in Libya shows that the adoption rate of MAIs in the manufacturing sector is higher than in the non-manufacturing sector. This finding also reveals that although the adoption rate of MAIs is low, it is still higher than the adoption rate in other studies in the same area that were undertaken in Libya earlier. ABC, benchmarking, and Kaizen have the highest adoption rate in both sectors, although the manufacturing sector comes first in terms of the adoption rate. ABC has the highest adoption rate, with 43.2% in the manufacturing sector and 20.3% in the non-manufacturing sector.

The reason for the discrepancy in the adoption of MAIs between both sectors came from the idea that using these techniques in the manufacturing sector to calculate the cost of the products is seen as more important than calculating the cost of services in the non-manufacturing sector. Moreover, the importance of using MAIs in the non-manufacturing sector is lower for two reasons: The first reason is that Libya is one of the less developed countries, and its economy is based mainly on revenues from exporting oil. This situation has led to a weak economy that lacks competition and productivity. The second reason is that management accounting in Libya is in its early stages, focusing on traditional MAPs in the manufacturing sector and ignoring the non-manufacturing sector.

Hypothesis 2:

There are no significant differences among contextual factors (contingent and institutional) in terms of their level of associations with MAIs in the manufacturing and non-manufacturing sectors.

In this part, the respondents were asked to specify the importance of each factor in terms of facilitating the adoption process by choosing one of the values on a five-point Likert scale sorted in order from 1 = do not facilitate to 5 = extremely facilitate.

Table 16 shows the top ten factors that may facilitate the adoption of MAIs in the manufacturing sector, listing the factors' importance and ranking according to their mean. After reviewing the relevant literature and similar studies conducted in the same area, 21 factors were chosen to assess the most influential factors that may facilitate adopting MAIs in Libya. These factors are contingent (13) factors and institutional (8) factors. The discussion related to Table 16 is based on whether the nature of each factor belongs to contingency or institutional theory to assess the influence of both theories on the adoption of MAIs.

**Table 16.** The top ten factors that may facilitate the adoption of MAIs in the manufacturing sector.

| Factor | Rank | Classification |
|---|---|---|
| Using computer systems for MA purposes | 1 | Contingency |
| Top management support | 2 | Contingency |
| Management accounting training programmes | 3 | Institutional |
| Accounting education in Libya | 4 | Institutional |
| The arrival of new accountants | 5 | Contingency |
| Production technology | 6 | Contingency |
| The authority attributed to the accounting function | 7 | Contingency |
| The availability of adequate accounting staff | 8 | Contingency |
| The competitiveness of the market | 9 | Contingency |
| Headquarters and government regulation | 10 | Institutional |

To assess the influence of the factors that belong to contingency and institutional theories on the adoption of MAIs within the Libyan manufacturing sector, the factors were divided into two groups, as shown in Table 16 (which contains contingent and institutional factors, sorted in order based on their influence on adopting MAIs). It can be seen from Table 16 that there are seven factors related to contingency theory, which have a significant influence on adopting MAIs, while there are only three factors related to institutional theory. This result provides the implication that the contingency factors are the most influential facilitators in the adoption and implementation process.

Table 17 shows the top ten factors (out of twenty-one factors included in this study) that may facilitate the adoption of MAIs in the non-manufacturing sector. These factors are in order from the top ranked, 1, to the lowest ranked, 10, according to their mean. Twenty-one factors were chosen to assess the most influential factors that may facilitate the adoption of MAIs in Libya. These factors are 13 contingent factors and 8 institutional factors. The discussion related to Table 17 is based on whether the nature of each factor belongs to contingency or institutional theory to assess the influence of both theories on adopting MAIs.

**Table 17.** The top ten factors that may facilitate the adoption of MAIs in the non-manufacturing sector.

| Factor | Rank | Classification |
|---|---|---|
| Using computer systems for MA purposes | 1 | Contingency |
| Top management support | 2 | Contingency |
| The availability of adequate accounting staff | 3 | Contingency |
| Management accounting training programmes | 4 | Institutional |
| The arrival of new accountants | 5 | Contingency |
| The authority attributed to the accounting function | 6 | Contingency |
| Adequate financial resources | 7 | Contingency |
| Headquarters and government regulation | 8 | Institutional |
| The competitiveness of the market | 9 | Contingency |
| Accounting education in Libya | 10 | Institutional |

We can notice from Table 17 that contingency factors have a stronger influence (than institutional factors) on adopting MAIs in the non-manufacturing sector. Seven contingency factors are among the top ten factors, as shown in Table 17, while institutional factors have only three factors perceived to be among the top ten influential factors in the adoption process.

Comparing the two tables indicates that nine factors are among the top ten factors in both Tables 16 and 17, with some differences in their ranking between the two tables, which means that these factors have an essential influence in the manufacturing and non-manufacturing sectors. These factors are as follows: using computer systems for MA purposes, top management support, management accounting training programmes,

accounting education in Libya, the arrival of new accountants, the authority attributed to the accounting function, the availability of adequate accounting staff, the competitiveness of the market, and headquarters and government regulation.

Using computer systems for MA purposes came first in both sectors, followed by top management support that came second in both sectors, which reflects the high importance of these two factors. On the one hand, the production technology factor ranked sixth in the manufacturing sector, whereas it was not among the top ten factors in the non-manufacturing list, and this could be due to its great importance in the manufacturing sector as it is part of the production process and used technology. On the other hand, the adequate financial resources factor ranked seventh among the top ten factors in the non-manufacturing sector, while it was not among the top ten factors in the manufacturing sector list.

Table 18 summarises the result of all of the hypothesis testing.

**Table 18.** *t*-Test results.

| Hypothesis 1 | MAIs | *t*-Test Result |
|---|---|---|
| | ABC | There is statistical sig difference between the two sectors |
| | ABM | There is no statistical sig difference between the two sectors |
| | BSC | There is no statistical sig difference between the two sectors |
| | TC | There is no statistical sig difference between the two sectors |
| | LCC | There is no statistical sig difference between the two sectors |
| | Benchmarking | There is no statistical sig difference between the two sectors |
| | Kaizen | There is no statistical sig difference between the two sectors |

MAIs in this study included seven advanced MAPs, namely, ABC, ABM, BSC, TC, life-cycle costing, benchmarking, and Kaizen. The results show that the adoption rate of MAIs is higher than the adoption rates of MAIs in previous studies conducted in the Libyan context (e.g., Leftesi 2008; Alkizza 2006; Abugalia 2011; Abulghasim 2006).

Table 19 shows the seven MAIs in the manufacturing and non-manufacturing sectors ranked according to their adoption rate. The ABC technique came first in the manufacturing and non-manufacturing sectors, followed by benchmarking, which came second in both sectors. Kaizen occupied third place in the manufacturing and non-manufacturing sectors, while target Costing came fourth in the manufacturing sector, occupying fifth place in the non-manufacturing sector. Similarly, ABM came fifth in the manufacturing sector. However, it occupied sixth place in the non-manufacturing sector. Life-cycle costing occupied sixth place in the manufacturing sector and came fourth in the non-manufacturing sector. Finally, BSC came seventh in both sectors.

**Table 19.** The adoption rate of MAIs.

| Technique | Manufacturing Sector | | Technique | Non-Manufacturing Sector | |
|---|---|---|---|---|---|
| | N | Adoption Rate | | N | Adoption Rate |
| ABC | 44 | 43.2 | ABC | 59 | 20.3 |
| Benchmarking | 44 | 27.3 | Benchmarking | 59 | 18.6 |
| Kaizen costing | 44 | 27.3 | Kaizen costing | 59 | 16.9 |
| TC | 44 | 20.5 | LCC | 59 | 11.9 |
| ABM | 44 | 11.4 | TC | 59 | 10.2 |
| LCC | 44 | 11.4 | ABM | 59 | 10.2 |
| BSC | 44 | 2.3 | BSC | 59 | 5.1 |

From Table 19, apart from ABC, which has a relatively high adoption rate in the manufacturing sector, the rest of the MAIs have a low adoption rate. Moreover, the adoption rate of MAIs in the manufacturing sector is higher than that in the non-manufacturing sector.

## 5. Result and Discussion

The results show that the adoption rate of MAIs is lower than the adoption rate of TMAPs; however, the adoption rate is considered to be higher than the adoption rates of MAIs in previous studies conducted in the Libyan context such as those from (Leftesi 2008; Alkizza 2006; Abugalia 2011; Abulghasim 2006).

The seven MAIs were ranked according to their mean value. The ABC technique came first, with a mean value = 3.69. It was adopted by 30.1% of respondent organisations and used as a trial by 5.8% of respondent organisations. Kaizen costing came second, with a mean value = 2.84 and an adoption rate of 22.3%, and was used as a trial by 5.8%, followed by benchmarking in third place, with a 21.4% adoption rate and 21.4% of respondent organisations using it as a trial. Target costing came fourth, with a 14.6% adoption rate and 4.9% of respondent organisations using it as a trial. Life-cycle costing, balanced scorecard, and activity-based management occupied the fifth to seventh places, with mean values of 2.50, 2.47, and 2.44, respectively.

In brief, the main findings related to the first question show that most TMAPs were in use in Libyan manufacturing companies. Although the expectations of adopting MAIs were low, the adoption rate of MAIs in this study indicates that it is higher than that in previous studies in the Libyan environment.

One reason for the low adoption rate of MAIs is the ownership type, where state-owned or recently privatised former state-owned companies represent an important percentage of the surveyed companies. Companies working under governmental control usually do not seek profit or competition; however, they must achieve social and political objectives. Therefore, these types of companies focus on complying with regulations and state finance law. This also might explain the high adoption rate of budget MAPs as their adoption is imposed by state regulations.

The second reason for the low adoption rate of MAIs in Libya is that the manufacturing and non-manufacturing industries in Libya are still in their early stages; therefore, they do not use sophisticated processes when they are carrying out their jobs. Accordingly, the level of benefits obtained from TMAPs is high, and the expected benefits that might be obtained from adopting MAIs is deemed low.

The third reason behind the low adoption of MAIs is the current unstable situation in Libya since February 2011, and the lack of economic, political, and social security.

The second question of this study is about the main factors that may hinder and/or facilitate the adoption of MAIs. The focus is on the role of contingency factors, institutional factors, and a combination of contingency and institutional factors.

The framework adopted in this study comprises two theories: the first theory is contingency theory, whereas the second theory in the framework is NIS. Seven dependent factors were chosen from the relevant literature as influencing MAIs in this study. The collective influence of each group of independent factors on each dependent variable of MAIs was tested by formulating hypotheses. These hypotheses were tested by using SPSS software to conduct multiple regression tests. The summary of the multiple regression tests is as follows.

*Contingency factors:* This group comprises thirteen independent factors, and the multiple regression aimed to examine the influence of this group on seven dependent factors in terms of the adoption process. The result shows that the independent factors have a strong impact on adopting four MAIs, namely, ABC, ABM, BSC, and Kaizen. The most influential factors that led to the adoption of these four MAIs were using a computer system for MA purposes, the loss of market share, the competitiveness of the market, the arrival of a competent accountant, adequate financial resources, and production technology.

*Institutional factors:* This group comprises eight independent factors, and the result shows that these factors facilitate adopting four out of seven MAIs in this study, namely, ABC, ABM, BSC, and benchmarking. The most influential variables reported in this group were MA training in Libya, specialist MA journals, headquarters and governmental regula-

tions, professional accounting bodies in Libya, and conferences, seminars, consultations, and workshops.

*A combination of contingency and institutional factors:* This group comprises 21 independent factors, and these had an impact on adopting five MAIs, namely, ABC, ABM, BSC, benchmarking, and Kaizen. Ten variables were considered as having the highest impact on adopting MAIs, namely, specialist MA journals, MA training programmes in Libya, using a computer system for MA purposes, headquarters and governmental regulations, professional accounting bodies in Libya, the competitiveness of the market, adequate financial resources, the arrival of competent accountants, production technology, and deterioration in profitability.

On the other hand, regarding factors that might impede the adoption of MAIs, the questionnaire form contained 21 factors selected from the relevant literature related to MA change and the diffusion of innovations. The result of the descriptive analysis of the questionnaire indicates that among the top ten factors considered as the most impeding, there were eight factors from contingency theory and two factors from institutional theory.

The top ten factors that may hinder the adoption of MAIs, sorted in order from high to low influence (see Table 20), are as follows.

**Table 20.** The top ten factors that may hinder the adoption of MAIs

| Factor | Mean | Rank | Classification |
|---|---|---|---|
| Lack of skilled employees | 4.17 | 1 | Contingency |
| Lack of local training programmes in MAIs | 4.11 | 2 | Institutional |
| Lack of support from top management | 4.03 | 3 | Contingency |
| Lack of software packages relevant to MAIs | 3.99 | 4 | Contingency |
| Lack of courses related to MAIs in academic institutions | 3.89 | 5 | Institutional |
| Lack of employee awareness of the benefits of MAIs | 3.81 | 6 | Contingency |
| Lack of confidence in the value of MAIs | 3.70 | 7 | Contingency |
| Lack of the competitiveness of the market | 3.60 | 8 | Contingency |
| Centralisation | 3.56 | 9 | Contingency |
| Lack of trust in change | 3.47 | 10 | Contingency |

Meanwhile, the factors that have the lowest influence in terms of hindering the adoption of MAIs (see Table 21) are as follows.

**Table 21.** The factors with the lowest influence in terms of hindering the adoption of MAIs

| Factor | Mean | Rank | Classification |
|---|---|---|---|
| Lack of co-operation between universities (academics) and companies (professionals) | 3.09 | 17 | Institutional |
| Lack of up to date publications about MAIs | 2.97 | 18 | Institutional |
| Lack of an active MA society | 2.94 | 19 | Institutional |
| Complexity of MAIs | 2.81 | 20 | Contingency |
| High operational cost of MAIs | 2.79 | 21 | Contingency |

## 6. Conclusions and Contribution

The results indicate that the adoption of ABC varies significantly between the manufacturing and non-manufacturing sectors. However, the results show no statistically significant difference in terms of the adoption of the other six MAIs between the manufacturing and non-manufacturing sectors.

Regarding factors influencing the adoption of MAIs, this study employed two groups/types of factors, namely, contingency and institutional factors. Contingency factors comprised thirteen independent factors, and institutional factors comprised eight factors. The results indicate that the most facilitating factors were contingency factors in both sectors. In the manufacturing sector, on the top ten list, there are seven factors related to contingency theory that have a significant influence on adopting MAIs, while there are only three

factors related to institutional theory. This result provides the implication that contingency factors are the most influential facilitators in the adoption and implementation process. Similarly, in the non-manufacturing sector, contingency factors have a stronger influence than institutional factors in the adoption of MAIs, where seven contingency factors are among the top ten, while only three institutional factors are perceived to be among the top ten.

Using computer systems for MA purposes came first in both sectors, followed by top management support in both sectors, which reflects the significance of these two factors. The production technology factor ranked sixth in factors that influence the adoption of MAIs in the manufacturing sector, whereas it was not among the top ten factors in the non-manufacturing list, while the adequate financial resources factor ranked seventh among the top ten factors that may facilitate the adoption of MAIs in the non-manufacturing sector, whereas it was not among the top ten factors in the manufacturing sector list.

This study provides several theoretical contributions to the literature on MA and the adoption of innovations. Moreover, this research is one of the first attempts to understand and explain the factors that influence the adoption of MAIs in Libya as follows.

*Firstly*, this study contributes to the MA literature of less developed countries in general and Libya in particular, and accordingly, it fills part of the gap in the extant literature and paves the way for future studies on MA based on the results of this study.

*Secondly,* this study employed triangulation in a theoretical framework that contains two different theories, contingency and NIS. Therefore, this study provides a good practical example of combining two theoretical approaches to gain a better understanding of a problem under study than a mono approach.

*Thirdly,* reviewing the relevant literature showed that the majority of previous studies undertaken in developing countries focused on describing and reporting the status of the adoption rate of TMAPs. This study covered manufacturing and non-manufacturing companies and tested the influence of 21 different independent variables on the adoption of MAIs.

*Fourthly,* the important contribution to the body of knowledge is the ability to employ and combine contingency and NIS theories in one study, in addition to adopting factors used by Innes and Mitchell (1990) in their study. NIS is convenient in explaining the external and environmental factors that may affect an organisation. It adopts three mechanisms (coercive, normative, and mimetic). Contingency theory is suitable to test the environmental change and uncertainty, work technology, and the size of a company as factors that may influence the adoption of MAIs. Thus, one more distinguishing attribute of this study is that it involved a larger number of independent variables than any other study conducted in Libya.

*Fifthly,* using a list of contingent and institutional factors that influence the adoption of MAIs provides a good foundation for future studies to conduct comparative or replicated studies to confirm or provide more insights into the factors that affect the adoption of MAIs.

*Sixthly,* this study combined two theories and developed a framework based on these two theories. This framework was used to investigate and explain the factors that may facilitate or hinder the adoption of MAIs in Libya. Although this framework did not fully explain the adoption of MAIs in Libya, the findings were satisfactory, and more theories need to be tested to gain a full explanation of the adoption process in Libya. In summary, the study framework represents one of the most important contributions of this study.

From a practical perspective, this study has several contributions to the practice regarding the adoption of MAIs as follows.

*Firstly,* this study covered manufacturing and non-manufacturing companies, testing the influence of 21 different independent variables on the MAI adoption process. Moreover, this study raised issues that have not been discussed in previous studies.

*Secondly,* the deep analysis, the classification of the independent variables, the variety of industries covered in this study, and the number of chosen MAIs as dependent vari-

ables mean the findings of this study represent an important contribution to the body of knowledge in the Libyan environment. Accordingly, this study is a step forward to help in tackling all impediments that may prevent local organisations from adopting MAIs.

*Thirdly,* the data used in this study were primary data collected by the researchers themselves using questionnaires. Even though most of the questions in the questionnaire form were chosen from the relevant literature, some of them were modified to serve the purpose of the questionnaire and to be relevant to all sectors included in this study. More precisely, this study provides a better explanation of factors that influence the adoption of MAIs in Libya because it covers different sectors and uses the findings of the interviews to support and complement the findings of the questionnaires.

*Fourthly,* factors that may influence the adoption of MAIs in Libya were classified into factors that may facilitate the adoption process, including 21 factors, and factors that may impede the adoption of MAIs in Libya, comprising 21 factors. Additionally, the researchers chose these variables to be analysed deeply in order to test the study's hypotheses. These variables were categorised into three main groups, namely, contingency variables, institutional variables, and a combination of contingency and institutional variables.

*Fifthly,* the findings of this study will be available and valuable to academics, professionals interested in MA, and governmental officials to help them to gain an overview of the adoption rate of TMAPs and MAIs and the factors that influence the adoption of MAIs within the Libyan environment. Additionally, this study can be used as a reference for decision makers in Libyan organisations to help them make suitable decisions.

**Author Contributions:** Literature review, A.B.; Methodology and design of questionnaire, A.B. and H.Y.; Data Collection, A.B.; Data analysis, A.B. and H.Y.; Discussion and writing ug, A.B. and H.Y. and D.A. All authors have read and agreed to the published version of the manuscript.

**Funding:** This research received no external funding.

**Conflicts of Interest:** The authors declare no conflict of interest.

## Appendix A. General Information

Information about the participant

| A1) Your job title: | | | | | |
|---|---|---|---|---|---|
| ☐ Financial accountant | ☐ Cost accountant | ☐ Management accountant | | | |
| ☐ Financial Manager | ☐ Internal auditor | Other, please specify.......................................... | | | |

| A2) work Experience: | <3 years | 3–5 years | 6–10 years | 11–15 years | >15 years |
|---|---|---|---|---|---|
| In this position | ☐ | ☐ | ☐ | ☐ | ☐ |
| In this organisation | ☐ | ☐ | ☐ | ☐ | ☐ |
| Overall experience | ☐ | ☐ | ☐ | ☐ | ☐ |

| A3) Gender & Age: | | | | |
|---|---|---|---|---|
| Gender | ☐ Male | ☐ Female | | |
| Age | ☐ <25 | ☐ 25–35 | ☐ 36–45 | ☐ >45 |

| A4) Participant's Academic qualification: | | | |
|---|---|---|---|
| ☐ High school level/Medium diploma | ☐ Bachelor/High institution | ☐ Master's | ☐ PHD |
| ☐ Professional qualification (e.g., CIMA, CPA, ACCA, CIPA) please indicate........................ | | | |
| A5) Participant's field of study: | | | |
| ☐ Accounting    ☐ Business administration | ☐ Economics | ☐ Finance | |
| Other, please specify.......................................................................... | | | |

Information about the organisation

| A6) The ownership |
|---|
| ☐ State owned Organisation (100% owned by the state). |
| ☐ Private organisation (100% owned by the private sector). |
| ☐ Mixed ownership between state and private sector.          State ownership.......% |
| ☐ Joint venture (ownership divided between the state and a foreign partner).     State ownership......% |
| If yes, when was the joint venture established?         ......................years ago. |
| ☐ Joint venture (ownership between private sector and a foreign partner). Private ownership...........% |
| If yes, when was the joint venture established?         ......................years ago. |

| A7) Is the business an independent company or a subsidiary company? |
|---|
| ☐ Independent company |
| ☐ Subsidiary company, Name of parent company (Optional) and % of their ownership..................... |
| A8) Type of business |
| ☐ Engineering      ☐ Food      ☐ Clothes      ☐ Oil and gas      ☐ Agriculture sector      ☐      Construction sector     ☐ Finance sector (including banking & insurance) ☐ Information technology sector (including telecommunication, telephone & internet) |
| ☐ Transport sector (including road, sea & air transport) |
| ☐ Commerce sector (including retail, wholesale and import & export trading) |
| ☐ Hotel      ☐ restaurant      ☐ travel      ☐ entertainment      ☐ professional services |
| Other, please specify......................................................... |

| A9) Number of years the organisation has operated: |
|---|
| ☐ Less than 5 years          ☐ 5–10 years          ☐ 11–15 years |
| ☐ 16–20 years          ☐ More than 20 years |

| A10) Number of employees |
|---|
| ☐ Less than 50      ☐ 50–100      ☐ 101–200      ☐ 201–500      ☐ More than 500 |

| A11) Approx. organisation's revenues according to last financial statements (Million LD): |
|---|
| Total revenue      ☐ Less than 1      ☐ 1–5      ☐ 6–15      ☐ 16–30      ☐ More than 30 |

| A12) Is this organisation one of the organisations that privatised after 1990s?     Yes ☐      No ☐ |
|---|
| If the answer is (Yes), please answer the following questions: |
| When did the privatisation process occur?.............................. |
| Did the organisation's strategy and goals change after privatisation process?     Yes ☐      No ☐ |
| Did the organisation emerge management accounting function after privatisation process?     Yes ☐      No ☐ |
| Did the organisation develop and underpin the cost system after privatisation process?     Yes ☐      No ☐ |
| Did the organisation adopt any of management accounting innovations after privatisation process?     Yes☐ No ☐ |

## Appendix B. Management Accounting Practices in Use

| **B1) Please tick any of the following roles and departments that exist in your organisation** |
|---|
| ☐ Cost accountant          ☐ Cost accounting department |
| ☐ Management accountant          ☐ Management accounting department |
| ☐ Financial analyst          ☐ Finance department |
| ☐ If none please indicate which department is responsible for MA tasks such as: budgeting, product costing, and performance evaluation, etc. ............................................................................................................. |

**B2) Please choose which techniques are currently in use by ticking the appropriate box √**

| Techniques | Does your organisation use this technique? | | If yes, please indicate the importance of this technique to your organisation. | | | | |
| --- | --- | --- | --- | --- | --- | --- | --- |
| | No | Yes | Not Important 1 | Below Average 2 | Average 3 | Above Average 4 | Extremely Important 5 |
| **Costing systems:** | | | | | | | |
| Variable costing | | | | | | | |
| Full (absorption) costing | | | | | | | |
| Standard costing | | | | | | | |
| Other please specify A).................................. B)............................... | | | | | | | |
| **Budgeting and control** | | | | | | | |
| Sales budget | | | | | | | |
| Production budget | | | | | | | |
| Cash budget | | | | | | | |
| Direct materials budget | | | | | | | |
| Direct labour budget | | | | | | | |
| Overhead budget | | | | | | | |
| Master budget | | | | | | | |
| Capital budgeting | | | | | | | |
| Flexible budget | | | | | | | |
| Zero- based budget | | | | | | | |
| Other, please specify A)........................................... B)........................................... | | | | | | | |
| **Performance measurement/evaluation** | | | | | | | |
| Return on investment (ROI) | | | | | | | |
| Residual Income (RI) | | | | | | | |

**B2) Please choose which techniques are currently in use by ticking the appropriate box √**

| Techniques | Does your organisation use this technique? | | If yes, please indicate the importance of this technique to your organisation. | | | | |
| --- | --- | --- | --- | --- | --- | --- | --- |
| | No | Yes | Not Important 1 | Below Average 2 | Average 3 | Above Average 4 | Extremely Important 5 |
| Economic value added (EVA) | | | | | | | |
| The share price | | | | | | | |
| Division profit | | | | | | | |
| Customer satisfaction | | | | | | | |
| Budget variance analysis | | | | | | | |
| Employees satisfaction | | | | | | | |
| Meeting budget target | | | | | | | |
| Other, please specify A).............................................. B).............................................. C).............................................. | | | | | | | |
| **Capital investment appraisal technique** | | | | | | | |
| Payback period | | | | | | | |
| Net Present Value (NPV) | | | | | | | |
| Internal Return Rate (IRR) | | | | | | | |
| Meeting the budget | | | | | | | |
| Accounting Rate of Return (ARR) | | | | | | | |
| Other, please specify A).............................................. B).............................................. | | | | | | | |
| **Decision support systems** | | | | | | | |
| Cost-volume-profit analysis | | | | | | | |
| Product life-cycle analysis | | | | | | | |

**B2) Please choose which techniques are currently in use by ticking the appropriate box √**

| Techniques | Does your organisation use this technique? | | If yes, please indicate the importance of this technique to your organisation. | | | | |
| | No | Yes | Not Important 1 | Below Average 2 | Average 3 | Above Average 4 | Extremely Important 5 |
|---|---|---|---|---|---|---|---|
| Product profitability analysis | | | | | | | |
| Sensitivity analysis | | | | | | | |
| Customer profitability analysis | | | | | | | |
| Other, please specify A)............................................ B)........................................... C)........................................... | | | | | | | |

**B3) Management accounting innovations (MAIs) in use**

Please tick one of the following statements which best describe the status of management accounting innovations (MAIs) in your organisation listed in the table below:

- **Never heard of it**: We are not familiar with this technique.
- **Never considered to adoption**: We are familiar with this technique but have not considered adoption.
- **Considered then rejected**: The technique has been evaluated then rejected.
- **Under consideration and as a trial**: Technique is under evaluation; however, the implementation decision has not been taken.
- **Currently used**: Technique was evaluated, approved and is in use now.

| M12Technique | Never heard of it | Never considered to adopt | Considered then rejected | Under consideration | Currently In use | |
| | | | | | As a trial | Fully implemented |
|---|---|---|---|---|---|---|
| Activity-Based Costing (ABC) | | | | | | |
| Activity-Based Management (ABM) | | | | | | |
| Balanced Scorecard (BSC) | | | | | | |
| Target Costing (TC) | | | | | | |
| Life-cycle costing | | | | | | |
| Benchmarking | | | | | | |
| Kaizen costing | | | | | | |
| Other, please specify A)............................. B)............................. C)............................. | | | | | | |

## Appendix C. Factors Influencing the Adoption of MAIs

(1)  Factors which facilitate the adoption of MAIs

**C1) Please indicate to what extent do the factors below facilitate the adoption of MAIs process**

| Factor | Do not facilitate 1 | Slightly facilitate 2 | Moderately facilitate 3 | Significantly facilitate 4 | Extremely facilitate 5 |
|---|---|---|---|---|---|
| The availability of adequate accounting staff | | | | | |
| Using computer systems for MA purposes | | | | | |
| The authority attributed to the accounting function within the organization | | | | | |
| The competitiveness of the market | | | | | |
| Production technology | | | | | |
| Product cost structure | | | | | |
| The loss of market share | | | | | |
| The arrival of a new accountant | | | | | |
| Deterioration in profitability | | | | | |
| Joint venture with foreign companies | | | | | |
| Top management support | | | | | |
| Conferences, seminars, consultations, and workshops | | | | | |
| Co-operation between universities (academics) and companies (professionals) | | | | | |
| Accounting research in Libya | | | | | |

**C1) Please indicate to what extent do the factors below facilitate the adoption of MAIs process**

| Factor | Do not facilitate 1 | Slightly facilitate 2 | Moderately facilitate 3 | Significantly facilitate 4 | Extremely facilitate 5 |
|---|---|---|---|---|---|
| Accounting education in Libya | | | | | |
| Management accounting training programmes | | | | | |
| Adequate financial resources | | | | | |
| Professional accounting bodies in Libya | | | | | |
| Specialist Management accounting journals | | | | | |
| Other, please specify A).......................... B).......................... C).......................... | | | | | |

(2)   Factors impeding the adoption of MAIs

**C2) Please indicate to what extent do the below factors impede the adoption of MAIs process.**

| Factor | Do not impede 1 | Slightly impede 2 | Moderately impede 3 | Significantly impede 4 | Extremely impede 5 |
|---|---|---|---|---|---|
| Lack of courses related to MAIs in academic institutions. | | | | | |
| Lack of local training programmes in MAIs | | | | | |
| Lack of financial resources | | | | | |
| Lack of skilled employees | | | | | |
| Lack of decision-making autonomy at lower levels | | | | | |
| Lack of compatibility between MAIs and the existing system | | | | | |
| Lack of an active MA society | | | | | |

**C2) Please indicate to what extent do the below factors impede the adoption of MAIs process.**

| Factor | Do not impede 1 | Slightly impede 2 | Moderately impede 3 | Significantly impede 4 | Extremely impede 5 |
|---|---|---|---|---|---|
| Lack of confidence in the value of MAIs | | | | | |
| Lack of up to date publications about MAIs | | | | | |
| Lack of support from top management | | | | | |
| Lack of software packages relevant to MAIs | | | | | |
| Lack of employee awareness of the benefits of MAIs | | | | | |
| Lack of foreign companies operating in Libya | | | | | |
| Lack of Libyan companies that have adopted MAIs | | | | | |
| Lack of co-operation between universities (academics) and companies (professionals) | | | | | |
| Lack of conferences, seminars and workshops about MAIs | | | | | |
| Lack of management accounting research in Libya | | | | | |
| Headquarters and government regulation | | | | | |
| Company ownership type | | | | | |
| Complexity of MAIs | | | | | |
| High cost of MAIs implementation | | | | | |

**C2) Please indicate to what extent do the below factors impede the adoption of MAIs process.**

| Factor | Do not impede 1 | Slightly impede 2 | Moderately impede 3 | Significantly impede 4 | Extremely impede 5 |
|---|---|---|---|---|---|
| Institutional Power | | | | | |
| Lack of trust in change | | | | | |
| Acceptance of routines | | | | | |

Thank you for your assistance in completing this questionnaire. If you have additional comments, please feel free give them in the space below.

.......................................................................................................................................................................................
.......................................................................................................................................................................................
.......................................................................................................................................................................................
.......................................................................................................................................................................................
.......................................................................................................................................................................................
.......................................................................................................................................................................................
.......................................................................................................................................................................................
.......................................................................................................................................................................................
.......................................................................................................................................................................................
.......................................................................................................................................................................................
.......................................................................................................................................................................................
.......................................................................................................................................................................................
.......................................................................................................................................................................................
.......................................................................................................................................................................................
.......................................................................................................................................................................................
.......................................................................................................................................................................................
.......................................................................................................................................................................................
.......................................................................................................................................................................

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
