# Peer review of "Contextual Factors and the Diffusion of MAIs in Manufacturing and Non-Manufacturing Sectors in Libya"

_jrfm, doi:10.3390/jrfm14090415_

Round 1

Reviewer 1 Report

Thank you for the opportunity to review this interesting article and especially the approach to Management Accounting Innovations. However, there are some suggestions that I would like to address to the authors as follows:

1. The abstract is much too long. Try to reduce it and synthesize it.

2. The literature should be divided so that readers can more easily understand the fluency of your ideas! In this sense, I ask the authors to divide the specialized literature on several sub-points, such as: 2.1. Management Accounting Innovations and the investigation stage, 2.2. Contextual factors addressed in this study, 2.3.1. Macro-context factors (Institutional / external factors), 2.3.2. Micro-organizational Factors (Contingent / Internal Factors), 2.4. The adopted framework or proposed framework.

3. The research methodology should be point number 3 in the article structure and number 4 Results or Main findigs and so on with numbering.

I found that some figures (figure 1) and tables (Table 2, Table 15) are not arranged properly so that they can be understood by readers and understood. Please correct this.

Author Response

Reviewer 1

Comments and Suggestions for Authors

Thank you for the opportunity to review this interesting article and especially the approach to Management Accounting Innovations. However, there are some suggestions that I would like to address to the authors as follows:

  1. 1. The abstract is much too long. Try to reduce it and synthesize it.

The abstract has been reduced.

  1. The literature should be divided so that readers can more easily understand the fluency of your ideas! In this sense, I ask the authors to divide the specialized literature on several sub-points, such as: 2.1. Management Accounting Innovations and the investigation stage, 2.2. Contextual factors addressed in this study, 2.3.1. Macro-context factors (Institutional / external factors), 2.3.2. Micro-organizational Factors (Contingent / Internal Factors), 2.4. The adopted framework or proposed framework.

The literature is divided into sub-points as it has been advised by the reviewer.

  1. The research methodology should be point number 3 in the article structure and number 4 Results or Main findigs and so on with numbering.

The paper is restructure according to this suggestion.

I found that some figures (figure 1) and tables (Table 2, Table 15) are not arranged properly so that they can be understood by readers and understood. Please correct this.

The figure and the tables are reviewed and the suitable corrections are done.

Reviewer 2 Report

The paper addresses an interesting topic. It´s main purpose is to analyses the extent of diffusion of Management Accounting Innovations (MAI) and to explore factors that influence the adoption of MAI by Libyan organizations’. So, this study contributes to MA literature of less developed countries.

The research seems to be well designed, and the methods employed are appropriate. The structure and methodology is adequate and data collected in a survey shows that some factors encourage the adoption of MAIs.

It is possible that there is a contribution worth of publishing in JRFM. However, authors need to carefully improve the paper to make this clear.

BROAD COMMENTS:

  1. Improve the abstract, why not to better refine it as a structured one?
  2. Improve introduction: Here motivation to conduct this survey is not well positioned with existing literature.
  3. There is a growing research literature on management accounting innovations (MAIs). The paper references need to be updated, and some new references should be added to the literature review section.
  4. Improve conclusion and future work. What are key take away messages and what interdisciplinary directions you see?

  1. Threats to validity need to be mentioned. How did you address some of those threats?
  2. Review article section numbering.

SPECIFIC COMMENTS

Abstract - Page 2 review the sentence: “The methodology of the study is based on a questionnaire.” A questionnaire is not a methodology, it´s a research instrument used to collect data. A survey is a research method.

Introduction –

Claims are made that are not properly supported in the literature. For example, on page 3: “This study was conducted in a less developed country to respond to different calls for undertaking further research that may overcome the limitations of the previous studies.” What calls for papers do the authors refer to? Which authors / studies suggest the lack of studies in this area?

Pages 4 and 5 - The relationship between research questions and gaps should be reviewed, adjusted and better corroborated in the literature.

The introduction section usually ends with the presentation of the article structure.

Literature review

The literature review must be deepened and updated (very few references are from the last 5 years).  A quick search shows some interesting recent articles. See for example:

Ax, C., & Greve, J. (2017). Adoption of management accounting innovations: Organizational culture compatibility and perceived outcomes. Management Accounting Research, 34: pp. 59-74. https://doi.org/10.1016/j.mar.2016.07.007

Busco, C., Caglio, A. and Scapens, R.W. (2015), Management and accounting innovations: reflecting on what they are and why they are adopted, Journal of Management and Governance, Vol. 19, No. 3, pp. 495-524.

Chenhall, R., & Moers, F. (2015). The role of innovation in the evolution of management accounting and its integration into management control. Accounting, Organizations and Society, 47, 1–13.

Chiwamit, P., Modell, S. and Scapens, R.W. (2017), Regulation and adaptation of management

accounting innovations: the case of economic value added in Thai state-owned enterprises,

Management Accounting Research, Vol. 37, pp. 30-48.

Johanson, D. and Madsen, D.Ø. (2019), Diffusion of management accounting innovations: a virus perspective, Journal of Accounting & Organizational Change, Vol. 15 No. 4, pp. 513-534. https://doi.org/10.1108/JAOC-11-2018-0121

Oyewo, B. (2021), "Do innovation attributes really drive the diffusion of management accounting innovations? Examination of factors determining usage intensity of strategic management accounting", Journal of Applied Accounting Research, Vol. 22 No. 3, pp. 507-538. https://doi.org/10.1108/JAAR-07-2020-0142

Zawawi, N.H.M. and Hoque, Z. (2010), Research in management accounting innovations: an overview of its recent development, Qualitative Research in Accounting and Management, Vol. 7 No. 4, pp. 505-568. https://doi.org/10.1108/11766091011094554

Research Methodology

Page 12 - Authors should explain here how the questionnaire was constructed and should add the document as an appendix.

Main findings section –

In this study the results are presented but not discussed. Therefore, a section should be added to discuss the results.

I hope these comments will be useful to improve your manuscript. I wish you all the best for your research.

Author Response

Reviewer 2

Comments and Suggestions for Authors

The paper addresses an interesting topic. It´s main purpose is to analyses the extent of diffusion of Management Accounting Innovations (MAI) and to explore factors that influence the adoption of MAI by Libyan organizations’. So, this study contributes to MA literature of less developed countries.

The research seems to be well designed, and the methods employed are appropriate. The structure and methodology is adequate and data collected in a survey shows that some factors encourage the adoption of MAIs.

It is possible that there is a contribution worth of publishing in JRFM. However, authors need to carefully improve the paper to make this clear.

BROAD COMMENTS:

  1. Improve the abstract, why not to better refine it as a structured one?

The abstract has been refined, reduced and improved.

  1. Improve introduction: Here motivation to conduct this survey is not well positioned with existing literature.

We added some limitations of the previous studies that covered by this study (p7).

  1. There is a growing research literature on management accounting innovations (MAIs). The paper references need to be updated, and some new references should be added to the literature review section.

Two new references have been added to the paper (p6 & p9)

  1. Improve conclusion and future work. What are key take away messages and what interdisciplinary directions you see?

  1. Threats to validity need to be mentioned. How did you address some of those threats?

Two paragraphs are added related to reliability and validity (section 4.1)

  1. Review article section numbering.

It has been revised and required correction is done

SPECIFIC COMMENTS

Abstract - Page 2 review the sentence: “The methodology of the study is based on a questionnaire.” A questionnaire is not a methodology, it´s a research instrument used to collect data. A survey is a research method.

It’s corrected and rewrite the sentence as the questionnaire is the data collection instrument.

Introduction –

Claims are made that are not properly supported in the literature. For example, on page 3: “This study was conducted in a less developed country to respond to different calls for undertaking further research that may overcome the limitations of the previous studies.” What calls for papers do the authors refer to? Which authors / studies suggest the lack of studies in this area?

This sentence is deleted as it is n’t necessary because the rationale behind conducting this study is presented in details in page 4.

Pages 4 and 5 - The relationship between research questions and gaps should be reviewed, adjusted and better corroborated in the literature.

After revision we found it’s properly explained and no additional explonations are needed.

The introduction section usually ends with the presentation of the article structure.

The structure of the paper is added (p4)

Literature review

The literature review must be deepened and updated (very few references are from the last 5 years).  A quick search shows some interesting recent articles. See for example:

The literature review has been updated by adding two new references as follows:

Ax, C., & Greve, J. (2017). Adoption of management accounting innovations: Organizational culture compatibility and perceived outcomes. Management Accounting Research, 34: pp. 59-74. https://doi.org/10.1016/j.mar.2016.07.007.

Oyewo, B. (2021), "Do innovation attributes really drive the diffusion of management accounting innovations? Examination of factors determining usage intensity of strategic management accounting", Journal of Applied Accounting Research, Vol. 22 No. 3, pp. 507-538. https://doi.org/10.1108/JAAR-07-2020-0142

Research Methodology

Page 12 - Authors should explain here how the questionnaire was constructed and should add the document as an appendix.

The questionnaire form has been added as an appendix.

Main findings section –

In this study the results are presented but not discussed. Therefore, a section should be added to discuss the results.

A section consists of discussion has been added to the paper (section 5, p30)
